# Micronutrients Affect Expression of Induced Resistance Genes in Hydroponically Grown Watermelon against *Fusarium* *o**x**y**sporum* f. sp. *niveum* and *Meloidogyne incognita*

**DOI:** 10.3390/pathogens11101136

**Published:** 2022-09-30

**Authors:** Kasmita Karki, Vishal Singh Negi, Tim Coolong, Aparna Petkar, Mihir Mandal, Chandrasekar Kousik, Ron Gitaitis, Abolfazl Hajihassani, Bhabesh Dutta

**Affiliations:** 1Department of Plant Pathology, University of Georgia, Tifton, GA 31793, USA; 2Department of Horticulture, University of Georgia, Athens, GA 30602, USA; 3Department of Plant Pathology, UC Davis, c/o U.S. Agricultural Research Station, Salinas, CA 93905, USA; 4Vegetable Laboratory, USDA, ARS, Charleston, SC 29414, USA; 5Department of Entomology and Nematology, Fort Lauderdale Research and Education Center, University of Florida, Institute of Food and Agricultural Sciences, Davie, FL 33314, USA

**Keywords:** induced resistance, watermelon, *Fusarium*, root-knot nematode, micronutrients

## Abstract

The soil-borne pathogens, particularly *Fusarium oxysporum* f. sp. *niveum* (FON) and southern root-knot nematode (RKN, *Meloidogyne incognita*) are the major threats to watermelon production in the southeastern United States. The role of soil micronutrients on induced resistance (IR) to plant diseases is well-documented in soil-based media. However, soil-based media do not allow us to determine the contribution of individual micronutrients in the induction of IR. In this manuscript, we utilized hydroponics-medium to assess the effect of controlled application of micronutrients, including iron (Fe), manganese (Mn), and zinc (Zn) on the expression of important IR genes (*PR1*, *PR5*, and *NPR1* from salicylic acid (SA) pathway, and *VSP*, *PDF*, and *LOX* genes from jasmonic acid (JA) pathway) in watermelon seedlings upon inoculation with either FON or RKN or both. A subset of micronutrient-treated plants was inoculated (on the eighth day of micronutrient application) with FON and RKN (single or mixed inoculation). The expression of the IR genes in treated and control samples was evaluated using qRT-PCR. Although, significant phenotypic differences were not observed with respect to the severity of wilt symptoms or RKN galling with any of the micronutrient treatments within the 30-day experimental period, differences in the induction of IR genes were considerably noticeable. However, the level of gene expression varied with sampling period, type and concentration of micronutrients applied, and pathogen inoculation. In the absence of pathogens, micronutrient applications on the seventh day, in general, downregulated the expression of the majority of the IR genes. However, pathogen inoculation preferentially either up- or down-regulated the expression levels of the IR genes at three days post-inoculation depending on the type and concentration of micronutrients. The results demonstrated here indicate that micronutrients in watermelon may potentially make watermelon plants susceptible to infection by FON and RKN. However, upon infection the IR genes are significantly up-regulated that they may potentially aid the prevention of further infection via SA- and JA-pathways. This is the first demonstration of the impact of micronutrients affecting IR in watermelon against FON and RKN infection.

## 1. Introduction

Watermelon (*Citrullus lanatus* var. *lanatus* (Thunb.) Matsum. & Nakai) is one of the most popular fruits in the world, with a total production of approximately 104 million tons [1]. The United States is the seventh-largest watermelon producer globally with 2.18 million tons of annual production and a farm gate value of USD 657 million [2]. The southeastern states including, Florida, Georgia, North Carolina, and South Carolina produce approximately 50% of the total national watermelon production in the United States. With 8903 hectares of watermelon cultivated area, Georgia is the third-largest watermelon producing state [2].

Watermelon diseases cause both economic and quality related losses in the Southern U.S. [3,4]. In recent years, soil-borne pathogens have become more prevalent due to the scarcity of methyl bromide (an effective soil fumigant) [5] and also partly because of limited land resources for rotation [6]. Fusarium wilt of watermelon, a disease caused by *Fusarium oxysporum* Schltdl.: Fr. f. sp. *niveum* (FON) W. C. Snyder and H. N. Hansen, is one of the most economically important diseases of watermelon worldwide [7]. Watermelon plant, once infected with FON, may develop symptoms, irrespective of its growth stage [8]. Under favorable conditions, FON infection may cause losses of up to 80% in yield and quality [9]. FON is a resilient pathogen and can form thick-walled chlamydospores. The chlamydospores aid FON to survive in soil as a saprophyte and on plant debris for multiple years [10,11]. FON isolates have previously been classified into four physiological races (zero, one, two, and three) based on their virulence on watermelon cultivars [12,13,14,15,16]. Among these four physiological races, FON race two has been reported to be widely distributed within the U.S., including Georgia. FON race three has been identified in Florida, Georgia, and Maryland [9,16,17]. Although widely cultivated seedless watermelon (triploid) cultivars exhibit some level of resistance to FON [18], they are highly susceptible to FON race three [16], making them a great threat to watermelon production. The rootstocks of interspecific hybrid squash (*Cucurbita maxima* Duch. Ex Lam. × *C. moschata* Duch. Ex Poir) and bottle gourd (*Lagenaria siceraria* (Molina) Standl.) are resistant to both FON races one and two [19,20,21] and therefore, grafting of susceptible watermelon scions onto such rootstocks can be a viable option. However, the use of rootstocks for growing susceptible watermelon is limited as grafted plants are considerably more expensive than non-grafted plants [20]. 

FON has also been reported to interact with southern root-knot nematode (RKN, *Meloidogyne incognita*) [8,22,23], which is an obligate parasite with a cosmopolitan distribution and a wide host range [24]. RKN induces galls on the roots of susceptible host and disrupts their vascular system resulting in poor growth and often leading to the death of the infected plants [25,26]. A survey in 2018 identified the presence of RKN in 50% of the watermelon fields in southern Georgia [27]. No commercially available watermelon cultivars are resistant to RKN, and yield losses have been predicted to reach approximately 20% in situations with high RKN populations [23,28,29]. Additionally, the interspecific hybrid squash and bottle gourd rootstocks used in grafted watermelons against FON are susceptible to RKN [30]. Recent reports indicate variable responses in terms of incidence and severity of wilt symptoms during FON and RKN interactions in watermelon [8,31,32,33]; further investigation is warranted to identify other factors that govern differential responses against FON and RKN. 

The strategies for disease management and sustainable production of crops usually include host resistance and induction of plant endogenous defense system. The currently available commercial watermelon cultivars lack resistance to both FON (race two and race three) and RKN, and therefore, host resistance does not appear to be feasible to deal with FON and RKN infection. However, the induction of plant endogenous defense systems by biotic and abiotic agents [34,35] offers a plausible alternative for disease management and sustainable production of watermelon. Induced resistance (IR) regulates the expression of defense genes by a variety of mechanisms including those dependent on SA- and JA-signaling. Simultaneous activation of the SA- and JA-mediated defense signaling pathways have been reported to induce systemic resistance (ISR) in watermelon against FON by *Bacillus velezensis* F21 [36]. The coordinated regulation of SA- and JA-signaling, along with redox signaling have been shown to increase resistance to RKN in watermelon [37]. 

SA and JA are crucial components in the pathogen- and wound-signaling pathways, which are often accompanied by induced expression of pathogenesis-related genes (*PR* genes) [38]. *PR* gene-encoded proteins have been reported to minimize pathogen populations and disease onset in non-infected plant parts of an infected plant [3]. In SA-dependent signaling, *PR1* and *PR5* are two important markers [39] and the non-expressor pathogenesis-related gene 1 (*NPR1*) is a global transcription factor [40,41], which is essential for regulating pathogenesis-related defense responses [42]. Plant defensins (*PDFs*), vegetative storage proteins (*VSPs*), and lipoxygenases (*LOXs*) are key genes involved in the JA-dependent signaling pathway [43,44,45]. The transcriptional activation of these genes represents a critical part of plants’ defense machinery against pathogens following pathogen or elicitor perception. 

Nutritional status, particularly micronutrients, serves as a factor affecting ISR in plants, and therefore, it can influence crop response to disease [46]. Micronutrients such as divalent metallic ions including iron (Fe), manganese (Mn), and zinc (Zn) are cofactors for a wide range of enzymes and are often essential for the activation of host defense responses following tissue infection [46]. The acquisition of metal ions by plants from their surrounding environment is pivotal for survival and defense against biotic and abiotic stress [47]. The presence of Fe, Mn, and Zn has been reported to promote disease resistance in the host plant [48,49,50]. The effect of soil micronutrients in plant defense genes against pathogens has been studied previously in a variety of crops. In *Alium cepa* (onions), the incidence of sour skin was found to be correlated with the soil and tissue Cu, Fe, Mn, and Zn concentrations [51]. The subsequent transcriptome analysis indicated upregulation (>5000 fold) of *PR1* in onion tissues collected from high soil Cu:Fe concentration ratios compared to low Cu:Fe concentration ratios [52]. A field study in *Nicotiana tabacum* (tobacco) also displayed a substantial correlation between tomato spotted wilt (TSW) and soil Cu:Fe values [53,54]. A risk model based on this study was developed [55], which successfully predicted TSW risk before planting in 2014 and 2015. Additionally, in a gene expression study, plants from a low-risk zone exhibited a 650-fold increase in the expression of *NPR1* compared to samples from a high-risk zone [55]. Similarly, another model can predict the risk of bacterial leaf spot (BLS) on *Capsicum annuum* (peppers) based on the micronutrient concentrations in the soil prior to pepper planting [54]. The defense genes from the SA pathway in peppers were significantly upregulated in predicted low-risk vs. predicted high-risk BLS sites. 

Although the above studies indicated the roles of soil micronutrients in promoting plant defense against pathogens, the specific roles of individual micronutrients in a complex-soil medium are still elusive. Experiments using soil-less medium (e.g., hydroponics) may aid in determining specific roles of individual micronutrients by reducing or negating the interactive effects that may often be present in a soil medium. Considering that FON and RKN are soil-borne pathogens, soil micronutrients may potentially play an important role in host-pathogen interactions during infection. However, limited information is available on this relationship. In this study, we used a hydroponic system to investigate whether Fe, Mn, and Zn (higher and lower than the standard dose) treatments can influence the expression of IR genes in the SA (*PR1*, *NPR1*, and *PR5*) and JA (*LOX*, *VSP*, and *PDF*) pathways in pre- and post-pathogen infection events (FON or RKN or FON+RKN) in watermelon.

## 2. Results

### 2.1. Mineral Content in Nutrient Solutions

The effects of micronutrients on IR against FON and/or RKN in watermelon were evaluated in greenhouse experiments using a hydroponic system (Figure 1). Three micronutrients (Fe, Mn, and Zn) were chosen in this study based on previous reports [56,57,58,59,60,61,62,63,64], indicating their potential role in host defense mechanisms. Watermelon seedlings received water as needed, and a single application of a specially formulated Steiner universal fertilizer solution (Table 1), which was modified from Steiner universal nutrient solution [65] with composition (ppm): N-168 (NH_4_H_2_PO_4_, KNO_3_, Ca(NO_3_)_2_), P-31 (NH_4_H_2_PO_4_), K-273 (KNO_3_), Ca-180 (Ca(NO_3_)_2_), Mg-48 (MgSO_4_), B-0.44 (H_3_BO_3_ ), Cu-0.02 (CuSO_4_), Mo-0.1 (Na_2_MoO_4_·2H_2_O), Fe-2 to 4 (Fe Chelate; Sequestrene 330), Mn-0.62 (MnCl_2_) and Zn-0.11 (ZnSO_4_·7H_2_O). ‘High’ and ‘low’ represent 3- and 0.5-times the concentration of the respective micronutrient in Steiner solution (Table 1). The concentrations of micronutrients (Fe, Mn, Zn) in hydroponic solutions were measured at 0- and 7-days post-treatment (DPT) for high, low, and Steiner solutions. In all the cases, the concentration of high-micronutrient was significantly (*p* < 0.05) highest followed by Steiner and high-micronutrient treatments, respectively at both 0 and 7 DPT (Table 2). 

### 2.2. Change in Micronutrient Level Influences the Expression of SA- and JA-Genes in Watermelon Seedlings

The transcript levels of six key genes—*pathogenesis-related gene 1* (*PR1*), *non-expressor pathogenesis-related gene 1* (*NPR1*), *pathogenesis-related gene 5* (*PR5*), *lipoxygenases* (*LOX*), *vegetative storage protein* (*VSP*), and *plant defensin* (*PDF*)—which are involved in plant IR, were determined using quantitative real-time reverse transcription-polymerase chain reaction (qRT-PCR). Transcripts were quantified in plants treated with micronutrient treatments at 7 DPT and compared with non-treated control plants in Steiner (Figure 2). Watermelon plants treated with high Fe for seven days significantly downregulated PDF, *PR1*, and *PR5* genes prior to inoculation. When high and low Fe treated plants were compared, relative expression of the *PR1* gene was significantly lower in plants treated with high Fe (Figure 2A). The expression of the *PR5* gene was significantly downregulated in low Fe treated plants but expression levels were not significantly different between high and low Fe (Figure 2A). Relative expression of *NPR1*, *PDF*, *PR1*, and *PR5* genes was significantly downregulated by the Mn treatment (Figure 2B). Relative expression of *PR1* and *VSP* genes was significantly higher for high Mn compared to low Mn-treated plants. The expression of *NPR1*, *PDF*, *PR1*, and *PR5* genes was significantly downregulated by Zn treatments (Figure 2C). When high and low Zn treated plants were compared, *VSP* gene expression was significantly higher in the low Zn treatment compared to the high Zn treatment (Figure 2C). Primer sequences and PCR conditions for test and reference genes are given in the Table 3.

### 2.3. Differential Expression of SA- and JA-Genes in Fe-Treated Plants Inoculated with FON, RKN, or Both

FON inoculated plants with high Fe and Steiner treatment displayed upregulation of *PDF* and *LOX* genes, respectively (Figure 3A). RKN inoculation resulted in significant downregulation of the *NPR1* gene in the plants treated with high Fe (Figure 3D). *NPR1* and *PR1* genes were downregulated in low Fe-treated plants with co-inoculation of both FON and RKN (Figure 3G). When Fe-treated plants were compared, expression of the *PDF* gene was higher in plants treated with high Fe with FON inoculation, and the same was true for the *PR1* and *VSP* genes after inoculation with both FON and RKN (Figure 3A,G).

### 2.4. Differential Expression of SA- and JA-Genes in Mn-Treated Plants Inoculated with FON, RKN, or Both

FON inoculation in plants treated with Steiner and high Mn resulted in the downregulation of *LOX* gene (Figure 3B). The *LOX* gene was also downregulated with RKN inoculation in plants treated with Steiner (Figure 3E). RKN inoculation significantly upregulated *VSP* gene expression in plants treated with low Mn (Figure 3E). When Mn-treated plants were compared, the relative expression of *LOX*, *PDF*, and *PR5* genes was higher in plants treated with low Mn with FON inoculation (Figure 3B). Inoculation of RKN resulted in higher expression of *LOX*, *NPR1*, and *VSP* genes in the same treatment (Figure 3E). After inoculation of FON and RKN, expression of PR1 and PR5 were again higher in plants treated with low Mn (Figure 3H).

### 2.5. Differential Expression of SA- and JA-Genes in Zn-Treated Plants Inoculated with FON, RKN, or Both

The *PR1* gene was significantly upregulated in plants treated with low Zn in response to co-inoculation with both FON and RKN (Figure 3I). Co-inoculation with both FON and RKN resulted in significant downregulation of the *PR5* gene in plants treated with high Zn (Figure 3I). When Zn-treated plants were compared, relative expression of the *LOX* gene was significantly higher in plants treated with high Zn compared to Steiner and low Zn with FON inoculation (Figure 3C). *LOX* and *PR5* gene expressions were higher in plants treated with high Zn after RKN inoculation (Figure 3F). However, co-inoculation with both FON and RKN reduced the expression of these genes in the high Zn treatment (Figure 3I).

### 2.6. FON Recovery and RKN Gall Rating

FON was successfully re-isolated from the tested plants that were previously FON-inoculated or co-inoculated with FON and RKN but not from the non-inoculated controls or the plants inoculated with only RKN. Isolates were also examined for FON using traditional PCR as previously described. The PCR assay successfully identified all of the purportedly isolated FON colonies from affected plants as FON. In plants colonized by FON, we found no noticeable difference in Fusarium wilt symptoms across micronutrient treatments. Furthermore, there were no differences in root galling between treatments. In contrast to non-inoculated plants or plants inoculated with FON, only plants that were inoculated with RKN or RKN coupled with FON exhibited root galling.

## 3. Discussion

The role of soil micronutrients on IR and plant diseases is well-documented in previous studies [52,53,54,55]. However, the findings from these studies were mostly based on soil-based media. Hence, it is often difficult to deduce what role these individual micronutrients play in IR during pre- or post-inoculation phases with plant pathogens. Additionally, little is known about the function of specific micronutrients in watermelon plant defensive reactions against soil-borne pathogens like FON and RKN. In this study, the influence of the level of micronutrients, Fe, Mn, and Zn, on the expression level of IR genes (*PR1*, *PR5*, and *NPR1* from SA-pathway, and *VSP*, *PDF*, and *LOX* from JA-pathway) in watermelon leaves pre- or post-inoculation with FON and RKN (single and mixed) in a hydroponics system were monitored through relative gene expression. In order to ascertain that the SA- and JA-genes are induced systemically away from the point of contact (roots), leaf samples (instead of root samples) were assayed for monitoring gene expression. The results indicate that micronutrient applications indeed induce SA- and JA-genes in leaves away from the point of contact (root).

The qRT-PCR results demonstrate significant differences in expression of IR genes among treatments and they varied with sampling period, type, and concentration of micro-nutrients applied, and pathogen-inoculation. We observed that plants treated with Fe, Mn, and Zn at higher and lower doses than those found in standard Steiner solution for seven days demonstrated downregulation of IR genes (*PR1*, *PR5*, *NPR1*, and *PDF*) (Figure 2A–C). The IR genes were not up-regulated even at 11 days in those treatments (Appendix A–C). However, upregulation was observed in some micronutrient-pathogen treatments at three days post-inoculation of the pathogen (equivalent to 11-day post micronutrient treatments). This result is in line with the previous study where no activation of the JA-pathway in the *Trichoderma hamatum* T382 pre-inoculated *Arabidopsis* plants was observed without a subsequent infection with *Botrytis cinera* [68]. 

At seven days post-treatment, none of the micronutrient treatments resulted in the upregulation of IR genes. However, following pathogen inoculation, tested genes in both SA- and JA-pathways were activated, and at three days post-inoculation (equivalent to 11-days post micronutrient treatments), their levels of expression varied substantially. These observations suggest that the type and concentration of micronutrients in watermelon may potentially influence SA or JA-mediated genes when infected with either FON or RKN or both. We did not observe significant phenotypic differences with respect to the severity of wilt symptoms or RKN galling among the micronutrient treatments within the 30-day experimental period. Although wilt symptoms were not observed, roots were colonized by FON as evident by the consistent recovery of the pathogen on the culture medium along with further confirmation with a PCR assay. It is possible that FON may need a soil system to induce wilt symptoms rather than a hydroponic system, as utilized in this study. Despite the lack of phenotypic differences, significant up- and down-regulation of IR genes among treatments were observed. It is possible that the induction of resistance genes (JA and SA pathways) might not be strong enough to induce phenotypic differences among different micronutrient treatments (type and concentrations). 

Although previous reports evaluated the expression of these genes in response to pathogens or micronutrients, none of the studies evaluated these variables in combination. In this study, we provide evidence that watermelon plants respond differentially to distinct micronutrient and pathogen combinations. For example, we observed upregulation in the following micronutrient-pathogen treatments: *PDF* gene in high Fe treated plants with FON inoculation, *VSP* gene in low Mn treated plants with RKN inoculation, and *PR1* gene in low Zn treated plants with co-inoculation of both FON and RKN. These observations indicate that genes in JA- (*PDF* and *VSP*) and SA-pathway (*PR1*) in watermelon plants respond differentially with respect to different micronutrients and pathogen combination treatments. The *PDF* and *VSP* genes are regulated by JA, a key compound of the JA-signaling pathway, and are induced during soil-borne pathogen infection [69,70]. In Bacillus-treated soybean, the upregulation of the *VSP* gene was observed upon inoculation with *Rhizoctonia solani* and *F.*
*oxysporum* [70]. The *VSP* is present in the host-vegetative tissues and has displayed mutagenic and phosphatase potential against herbivores [71,72]. Small, basic, cysteine-rich peptides known as plant defensins (PDFs) possess antibacterial activity against a variety of microbes [73,74]. The expression of *PDF1.2* and nine other genes was observed to be upregulated in *Arabidopsis thaliana* upon parasitic plant attack by *Orobanche ramose* [75].

Pathogenesis-related (PR) proteins are induced by products made in the SA-pathway and help defend plants against plant pathogens [76,77]. In the roots of soybean seedlings inoculated with *Phytophthora sojae*, the upregulation of *PR1* and *PR2* was connected to limited lesion development [78]. Similarly, high expression of defense-related genes, such as *PR1*, and greater activities of PR enzymes were found to increase watermelon’s resistance to FON in a wheat intercropping system [79].

Induced resistance mediated by several genes governing the SA- and JA-pathways can also be downregulated with specific micronutrient and pathogen combinations. We observed the downregulation of plant IR genes in response to inoculation with FON, RKN, or both. The *LOX* gene was downregulated in plants treated with Steiner only, and when grown with high Mn and inoculated with FON, and also in plants treated with the Steiner solution only but inoculated with RKN. The *NPR1* gene was downregulated in plants treated with high Fe after RKN inoculation and also in plants treated with low Fe in response to co-inoculation of FON and RKN. *PR1* gene was downregulated in the plants treated with low Fe in response to the co-inoculation with FON and RKN. Inoculation of FON and RKN significantly downregulated the expression of the *PR5* gene in plants treated with high Zn. The downregulation of these IR genes in response to the pathogens (FON or RKN or both) infecting watermelon in a particular micronutrient treatment may affect their defense response by enhancing host susceptibility.

In the current study, we discovered that high Fe-treated plant leaf tissue had the highest gene expression compared to low- and standard-Fe treated plants (Steiner). Several intrinsic host defense mechanisms require Fe [80]. In *Arabidopsis* [56,57] and wheat [58], cellular translocation of Fe to infection sites, which coincided with local reactive oxygen species (ROS) formation, revealed direct participation of Fe in the defense response. The biotrophic growth phase of *Colletptrichum graminicola* was found to be delayed and partially suppressed by maintaining appropriate Fe concentrations in maize [59]. Mn has been shown to induce the production of phenolic compounds and plant-IR mechanisms [60]. It is a cofactor of superoxide dismutase (SOD), which takes part in plant defense against oxidative stress brought on by an increase in the harmful ROS and reactive forms of oxygen (ROV) [61]. Effects of increased Mn availability on disease severity have been found to vary among different plant species with different diseases [62]. In our study, we observed that the relative expression of IR genes was higher in the plants treated with low Mn after pathogen inoculation. Relative expression of defense genes was higher with plants grown in the high Zn treatment in response to inoculation with either FON or RKN. However, expression was higher in plants treated with low Zn after co-inoculation of both FON and RKN. Zn treatment has been reported to reduce symptoms of disease in many cases [63,64,81]. The variability we observed in plants inoculated with FON and RKN might be due to the protective concentration of Zn against one pathogen might have induced increased susceptibility to another pathogen as reported previously [82]. 

Symptom differences were also not observed among treatments in terms of wilting and gall rating caused by RKN inoculation. The development of symptoms might have been affected by the hydroponics system where the plants had constant access to moisture as the roots were merged in the nutrient solutions.

## 4. Materials and Methods

### 4.1. Preparation of Inoculum of FON and RKN

An isolate of FON was collected from symptomatic watermelon plants in Georgia, U.S., and was identified as FON race two after inoculating them on a set of watermelon differentials as described previously [9,83]. A pure culture of FON was maintained on Potato Dextrose Agar (PDA) (Thermo Fisher Scientific, Waltham, MA, USA) at 25 °C for 7 days. Five mycelial plugs (0.7-cm diameter) were removed from the edge of the colony and were transferred to a 500 mL flask containing 200 mL of potato dextrose broth. Cultures were incubated at 25 °C and 160 rpm on a rotary incubator shaker (Thermo Scientific, Alachua, FL, USA). Microconidia were collected after 2 weeks by pouring the culture through double-layered sterile cheesecloth and using a hemocytometer; the concentration of the microconidia suspension was adjusted to a final concentration of 1 × 10^5^ microconidia/mL as described earlier [84]. A population of *M. incognita* race three (RKN) maintained on eggplant (*Solanum melongena* L.) was kindly provided by Dr. Richard Davis (USDA-ARS, Tifton, Georgia) and Nematology Lab (Tifton campus). Eggplants were maintained in a greenhouse at 22–30 °C. Plant roots that were infected were cut, cleaned in tap water, and then left in a mist chamber for 2–9 days to allow the eggs to develop. After incubation for 9 days, newly hatched second-stage juveniles (J2) were collected every 3 days. The J2s were also collected from the soil in which the culture was maintained by the sugar centrifugation method [85]. The resulting *M. incognita* (J2) collected were diluted in tap water to get a final concentration of 6000 J^2^/mL.

### 4.2. Hydroponics Set-Up under Greenhouse Conditions

Two independent greenhouse experiments were designed for use with a hydroponics system to evaluate the effects of micronutrients in watermelon on IR against FON or RKN or co-inoculation of FON and RKN. The greenhouse was maintained at 24 °C and 80% relative humidity throughout the experiment. A small layer of vermiculite was placed on top of the watermelon seeds (cv. Sugar Baby) after they were planted into sheets of 2.5 cm^2^/cell Rockwool cubes (Grodan Inc., Hedehusene, Denmark). Whenever needed, water was used to keep the sheet moist. Following germination, seedlings received water as needed, and a single application of a specially formulated Steiner universal fertilizer solution (Table 1). For this manuscript, we refer to “the modified Steiner universal nutrient solution” [65] as a “Steiner solution”. On Rockwool, seedlings were kept for three weeks after which, they were transferred to plastic containers (43 cm W × 30 cm D × 19 cm H, Reynolds Consumer Products, Inc., Louisville, KY, USA) with appropriate micronutrient treatments in addition to the Steiner solution (Table 1). High Fe (3X concentration of Fe in Steiner solution), low Fe (0.5X concentration of Fe in Steiner solution), high Mn (3X concentration of Mn in Steiner solution), low Mn (0.5X concentration of Mn in Steiner solution), high Zn (3X concentration of Zn in Steiner solution), low Zn (0.5X concentration of Zn in Steiner solution), and Steiner (X concentration of Fe (3 mg·L^−1^), Mn (1 mg·L^−1^) and Zn (0.4 mg·L^−1^)) were among the micronutrient treatments. All the micronutrient solutions were prepared from ACS-grade chemicals purchased from Sigma-Aldrich (St. Louis, MO, USA). The concentrations (mg. L^−1^) of individual ingredients are listed in Table 1. The micronutrient treatments were prepared in 8 L of deionized (dH_2_O) water in a separate plastic container. For the hydroponic system, a Styrofoam tray with holes at the bottom was used to facilitate direct access of roots to the micronutrient suspension. Three seedlings in rockwools were placed equidistantly on the Styrofoam plate, which was then placed on the top of the plastic container containing micronutrient treatment (Figure 1). A 15.24 cm aquarium air stone with plastic tubes (0.3 mm diameter) and an air pump (Pentair Aquatic Eco-Systems Inc., Apopka, FL, USA) were used to aerate the solutions (Figure 1). The volume of plastic containers was kept at its initial (8 L) level by adding dH_2_O every two days. Seedlings were treated with different micronutrient treatments for 7 days (Table 1). The micronutrient treatments were either infected with FON or RKN or co-inoculated with both at the 8-day post-treatment (DPT). A 1 ml suspension containing 1 × 10^5^ microconidia/mL was pipetted onto the rockwool cubes at the base of the watermelon seedling as part of the FON inoculation process. A 1 ml slurry containing 6000 active RKN J2s was pipetted similarly for nematode inoculation. A similar inoculation strategy was used and the inocula were applied concurrently for treatments that were inoculated with both FON and RKN.

Plants were maintained in nutrient solution for 23 days post-inoculation with a total experimental duration of 30 days (7 days pre-inoculation + 23 days post-inoculation) at 28 °C mean greenhouse temperature. The pH of the initial solutions was approximately 5.5 to 6 and did not change significantly during the experiment. The replicate of each micronutrient treatment was comprised of a plastic container with three seedlings each (Figure 1). Three replicates per treatment (a total of 9 seedlings per treatment) were used in an experiment and two independent experiments were conducted. A completely random design was used. The schematic depiction of the timeframe and treatments is shown in Figure 1.

### 4.3. Mineral Analysis

Samples of nutrient solution (20 mL) were collected from each container at 7-DPT and stored at 4 °C. Concentrations of Fe, Mn, and Zn (mg·L^−1^) were assessed using inductively coupled plasma atomic emission spectroscopy (ICP-AES) at the Waters Agricultural Laboratories (Camilla, GA, USA) for three replicates for each treatment.

### 4.4. Relative Expression of SA- and JA-Genes in Watermelon Seedlings Grown in Specific Micronutrient Solutions

In order to ascertain that the SA- and JA-genes are induced systemically away from the point of contact (roots), leaf samples instead of root samples were assayed for monitoring gene expression. Leaf samples were collected to determine the relative expression of SA- and JA-genes in watermelon seedlings grown in various micronutrient treatments before inoculation. Using a pair of sterile scissors (for each cut), 7-DPT, leaf samples (*n* = 3 per replicate/treatment) were obtained by cutting the third or fourth leaf from the apex of the stem/vine from hydroponically grown watermelon seedling under various treatments. Samples were snap-frozen using liquid nitrogen and then moved to a −80 °C freezer at the UGA Tifton Campus laboratory until they were required for additional analysis. Using the manufacturer’s instructions, total RNA was isolated from 100 mg of powdered leaf tissue after leaves were ground into a coarse powder under liquid nitrogen (RNeasy Plant Mini Kit, Qiagen, CA, USA) and NanoDropTM Lite (Thermo Scientific, Wilmington, DE, USA) was used to measure concentrations. The A260/A280 absorbance ratio, which was between 2.1 and 2.2 for all samples and indicated that the RNA was devoid of protein contamination, served as evidence of the RNA’s high quality [86]. 

For later usage, total RNA samples were aliquoted and kept at −80 °C. The iScript cDNA synthesis kit (Bio-Rad Laboratories Inc., Hercules, CA, USA) was used to reverse-transcribe 500 ng of total RNA to synthesize first-strand cDNA (Bio-Rad Laboratories Inc., Hercules, CA, USA). Utilizing SsoFast EvaGreen Supermix (Bio-Rad Laboratories Inc., Hercules, CA, USA), specific primer pairs for the marker genes, and a Smart Cycler System (Cepheid, Hercules, CA, USA), qRT-PCR was carried out (Table 3). The primers and qRT-PCR parameters for *PR1* and *PR5* genes were the same as described earlier [66]. The list of genes, primer sequences, and qRT-PCR parameters used in this study are listed in Table 3. These genes were selected based on their involvement in plant defense response mechanisms [68,69,70,71,72,75,76,77,78,79]. For real-time PCR, the resulting first-strand cDNA was diluted (1:10) and the specific genes were amplified from the diluted cDNA (5 ng) using SsoFast EvaGreen Supermix (Bio-Rad Laboratories Inc.) and specific primers. The *ß-Actin* gene was used as an internal control. The forward primer, 5′-CCATGTATGTTGCCATCCAG-3′, and reverse primer, 3′-GGATAGCATGGGGTAGAGCA-5′ were used for *ß-Actin* as described previously [67]. Plants in the Steiner treatment were considered as non-treated control. The 2^−∆∆CT^ method [87,88] was used to calculate the relative fold changes of the target genes. Relative gene expression was then compared for each treated (*n* = 48; 24 biological replicates and 2 technical replicates) and non-treated (*n* = 12; 6 biological replicates and 2 technical replicates) plants. 

### 4.5. Relative Expression of SA- and JA-Genes in Watermelon Seedlings, Grown in Specific Micronutrient Solutions and Inoculated with FON, RKN, or Both

The effects of micronutrient treatment and pathogen inoculation alone or in combination on the relative expression of SA- and JA-induction pathway genes in watermelon seedlings were investigated. Samples were harvested at 3 days post-inoculation (DPI) by cutting the third or fourth leaf from the apex of the stem with a pair of sterile scissors, yielding leaf samples (*n* = 3 per replicate/treatment) (equivalent to 11 DPT). Non-inoculated plants grown on the Steiner treatment were considered as the non-treated control. From each replicate, two technical replicates per sample were utilized for gene expression analysis. Further procedures spanning from the extraction of RNA to the determination of relative gene expression were performed in the same manner as mentioned above. 

### 4.6. FON Recovery and RKN Gall Rating

No visible symptoms of wilting in the plants were observed throughout the course of the experiment. In order to confirm the infection status, we used two approaches—(i) isolation of the pathogen from the inoculated and control samples, and (ii) PCR validation of infection in the inoculated and control samples. For the first approach, one plant from each container (*n* = 3 containers/treatment), either inoculated with FON alone or in combination with RKN and non-inoculated plants in the Steiner solution, was tested for the presence of FON at the conclusion of the experiment. Using a pair of sterile scissors, stem pieces were cut into 0.5-cm-long pieces from the base of the main stem of each plant. After being surface disinfested with 0.6% sodium hypochlorite for 1.5 min, the stem pieces were rinsed in sterile water, and then they were put on a semi-selective peptone pentachloronitrobenzene agar medium [89]. Following a 7-day incubation period at 25 °C, plates were examined under a microscope to identify fungal isolates based on morphological characteristics [90]. FON status was then validated using a PCR assay with the FON-specific primers Fon-1 and Fon-2 [91]. PCR confirmatory assays and morphological analysis were used to identify the percentage of plants that were infected with FON. The 10 cm of roots closest to the plant’s base that were inoculated with RKN were taken for gall assessment. Roots were washed after the removal of the rockwool plugs, and the number of galls was visually assessed using a 0–5 galling index (GI) as follows: 0–0 galls, 1–2 galls, 3–10 galls, 11–30 galls, 31–100 galls, and >100 galls [92]. 

### 4.7. Statistical Analysis

To normalize the data distribution before analysis, data were log-transformed (log10 (x + 1)) wherever appropriate. Untransformed arithmetic means are presented. Data from two trials were combined after ensuring no significant interaction of treatment was present at *p* ≤ 0.05. Experimental data were statistically analyzed using one-way Analysis of Variance (ANOVA) tool, and the Tukey–Kramer test was used to compare treatment effects at *p* < 0.05 level in SAS 9.4®.

## 5. Conclusions

The findings in this study suggest a possible association between the availability of specific nutrients and the induction of SA- and JA-genes in watermelon against FON and RKN. Importantly, the induction of SA- and JA-genes in leaves away from the point of contact (root) demonstrate the systemic induction of these genes. This also indicates a po-tential crosstalk between pathways affected by intracellular nutrient concentrations, in-cluding micronutrient homeostasis, and pathways governed by IR against FON and RKN infection, which may directly affect IR responses in watermelon. A detailed study is es-sential to understand how nutrient concentrations in substrate or leaf tissue of watermel-on affect the expression of IR genes, and also if FON and RKN induce the plant response by itself or in the micronutrient treatment-mediated manner.

## Figures and Tables

**Figure 1 pathogens-11-01136-f001:**
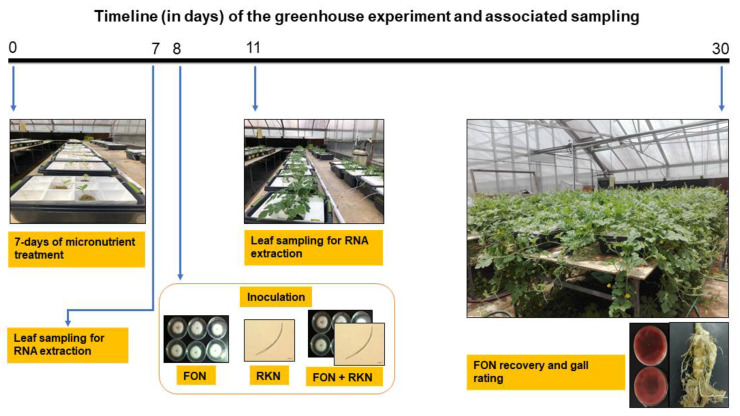
Schematic representation of the timeline (in days) of the greenhouse experiment and associated sampling to evaluate the effects of micronutrients (Fe or Mn or Zn) on induced resistance in watermelon seedlings when challenged with *Fusarium oxysporum* f. sp. *niveum* (FON) and *Meloidogyne incognita* (RKN) or both.

**Figure 2 pathogens-11-01136-f002:**
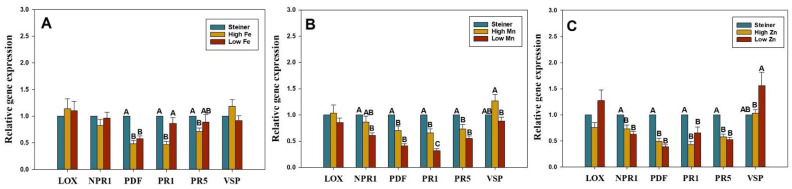
Relative expression of *non-expressor pathogenesis-related gene 1* (*NPR1*), *pathogenesis-related protein 1* (*PR1*), *pathogenesis-related protein 5* (*PR5*), *lipoxygenase* (*LOX*), *plant defensin* (*PDF*), and *vegetative storage protein* (*VSP*) genes by qRT-PCR in watermelon leaves at 7 days post treatment with micronutrients (**A**) Fe, (**B**) Mn, and (**C**) Zn. Watermelon seedlings (cv. Sugar Baby; 3 weeks old) were either treated with Fe or Mn or Zn at high (3X), low (0.5X), or standard concentration (X, Steiner) for 7 days. *ß-Actin* was used as the reference gene. Plants in the Steiner treatment were considered as non-treated control. Data are the mean fold changes ± SE in gene transcript levels in tissues from micronutrient treated plants relative to tissues from non-treated control plants in Steiner. Letters indicate a significant difference between treatments with the Tukey–Kramer test (*p* < 0.05).

**Figure 3 pathogens-11-01136-f003:**
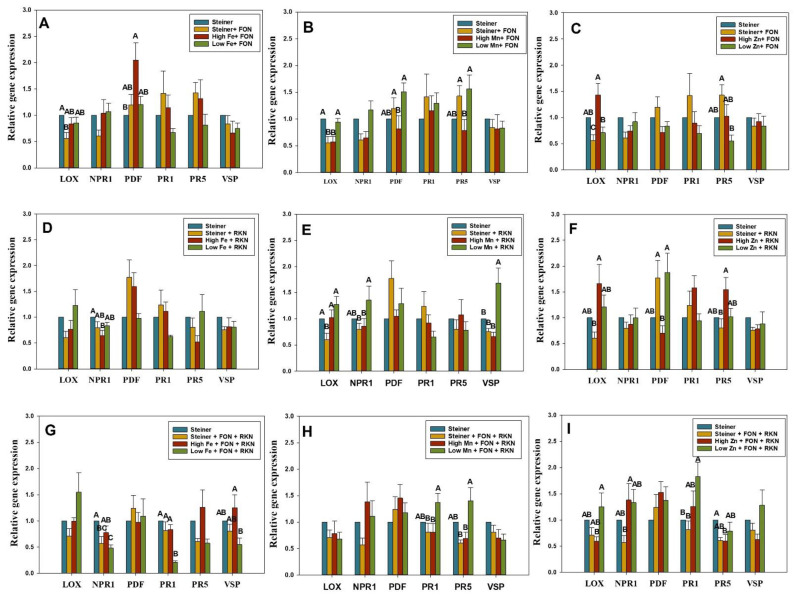
Relative expression of *non-expressor pathogenesis-related gene 1* (*NPR1*), *pathogenesis-related gene 1* (*PR1**)*, *pathogenesis-related gene 5* (*PR5**)*, *lipoxygenase* (*LOX**)*, *Plant defensin* (*PDF**)*, and *vegetative storage proteins* (*VSP**)* genes by qRT-PCR in leaves of watermelon plants at 11 days post-treatment with micronutrients Fe (**A**,**D**,**G**), Mn (**B**,**E**,**H**), and Zn (**C**,**F**,**I**) at high (3X), low (0.5X), and standard concentration (X) Steiner via hydroponics system and 3 days post-inoculation with 1 ml of 5 × 10^5^ microconidia of *Fusarium oxysporum* f. sp. *niveum* (FON) (**A**–**C**) or 1 mL of 6000 active J2s of *Meloidogyne incognita* (RKN) (**D**–**F**), or both (**G**–**I**). *ß-Actin* was used as the reference gene. Plants in the Steiner treatment were considered as non-treated control. Data are the mean fold changes ± SE in genes transcript levels of tissues from inoculated plants in micronutrient treatments relative to tissues from non-inoculated control plants in Steiner. Letters indicate a significant difference between treatments with the Tukey–Kramer test (*p* < 0.05). Primer sequences and PCR conditions for test and reference genes are given in Table 3.

**Table 1 pathogens-11-01136-t001:** Mineral concentrations (ppm) in nutrient solutions used in this study.

Micronutrient Treatments	N	P	K	Ca	Mg	B	Cu	Mo	Fe	Mn	Zn
Steiner ^a^	256	48	304	180	48	1	0.2	0.1	3	1	0.4
High Fe	256	48	304	180	48	1	0.2	0.1	**9**	1	0.4
Low Fe	256	48	304	180	48	1	0.2	0.1	**1.5**	1	0.4
High Mn	256	48	304	180	48	1	0.2	0.1	3	**3**	0.4
Low Mn	256	48	304	180	48	1	0.2	0.1	3	**0.5**	0.4
High Zn	256	48	304	180	48	1	0.2	0.1	3	1	**1.2**
Low Zn	256	48	304	180	48	1	0.2	0.1	3	1	**0.2**

^a^ Steiner solution is modified from Steiner universal nutrient solution with composition (ppm); N-168 (NH_4_H_2_PO_4_, KNO_3_, Ca(NO_3_)_2_), P-31 (NH_4_H_2_PO_4_), K-273 (KNO_3_), Ca-180 (Ca(NO_3_)_2_), Mg-48 (MgSO_4_), B-0.44 (H_3_BO_3_), Cu-0.02 (CuSO_4_), Mo-0.1 (Na_2_MoO_4_·2H_2_O), Fe-2 to 4 (Fe Chelate; Sequestrene 330), Mn-0.62 (MnCl_2_) and Zn-0.11 (ZnSO_4_·7H_2_O).

**Table 2 pathogens-11-01136-t002:** Mean concentrations (ppm) of Fe, Mn, and Zn in nutrient solutions. The concentrations of micronutrients (Fe, Mn, Zn) in hydroponic solutions were measured at 0- and 7-days post-treatment for high, low, and Steiner solutions.

Micronutrient	Micronutrient Treatment	Day 0 (ppm; Applied)	Day 7 (ppm)
Fe	High Fe	9.0 a^1^	9.28 ± 0.29 a
	Steiner	3.0 b	2.35 ± 0.33 b
	Low Fe	1.5 c	1.40 ± 0.05 c
Mn	High Mn	3.0 a	2.25 ± 0.12 a
	Steiner	1.0 b	0.57 ± 0.08 b
	Low Mn	0.5 c	0.29 ± 0.03 c
Zn	High Zn	1.2 a	1.03 ± 0.05 a
	Steiner	0.4 b	0.43 ± 0.08 b
	Low Zn	0.2 c	0.34 ± 0.07 b

^1^ Means ± standard error followed by different letters in same column under same section for each micronutrient are significantly different with the Tukey–Kramer test (*p* < 0.05).

**Table 3 pathogens-11-01136-t003:** The list of genes, primer sequences, and quantitative polymerase chain reaction (PCR) conditions used in this study.

Genes	Forward Primer Sequence (5′-3′)	Reverse Primer Sequence (3′-5′)	Comments	PCR Conditions
*NPR1*	CGCTGCCGATATGCATGTGA	GTCAACCTTCAGCAAGTTGCCA	This study	95 °C for 2 min; 35 cycles of 95 °C for 20 s, 62 °C for 30 s, and 72 °C for 60 s; and final extension of 72 °C for 6 min
*PR1*	GACTCGCCTCAAGACTTTGT	GATGCGTTGGTTGGCATATTG	[66]	95 °C for 3 min; 40 cycles of 95 °C for 10 s, 60 °C for 30 s, and 72 °C for 30 s; and final extension of 72 °C for 6 min
*PR5*	CCTGGAGCGTCAAAGTCATTTA	CTCCAGTTAAGCAGGTGATACG	[66]	same as above
*LOX*	TCTCAACTGTGCTCCCATTC	GGAAGCAGTGGCTTTGAATTAC	This study	same as above
*PDF*	GCGAAGGTGTGCGAGAA	CATGGCAAGCTCCATGTTTG	This study	same as above
*VSP*	ACCAAGGGAAGTCAGCAATAC	CCGAAACTGACGTACCCAATAA	This study	same as above
*ß-Actin*	CCATGTATGTGCCATCCAG	GGATAGCATGGGGTAGAGCA	[67]	[67]

## Data Availability

Not applicable.

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
