# Peer review of "Micronutrients Affect Expression of Induced Resistance Genes in Hydroponically Grown Watermelon against Fusarium oxysporum f. sp. niveum and Meloidogyne incognita"

_pathogens, 2022, doi:10.3390/pathogens11101136_

Round 1

Reviewer 1 Report

See attached.

Author Response

We are thankful for the valuable comments of the reviewer that helped us improve the manuscript. The response to the specific comments of the reviewer is added as a separate file. 

Reviewer 2 Report

The manuscript in reference describes the effect of controlled application of micronutrients in hydroponics (i.e., Fe, Mn, and Zn) on the expression of some induced resistance genes in watermelon seedlings inoculated with either Fusarium oxysporum f. sp. niveum (FON) and southern root-knot nematode Meloidogyne incognita, or both. The manuscript has relevant information and results that will be interesting for readers. However, some issues should be addressed prior to further consideration.

1.       Detailed scrutiny should be performed throughout the manuscript to revise/correct some grammar and stylistic issues.

2.       Abstract is too large. It can be reduced using the most important information.

3.       Lines 140-145: Revise text format. Other passages have the same mistake.

4.       Line 141: Specify the abbreviation DPT since it is the first time mentioned in the manuscript.

5.       Line 209: Figures and Tables should be embedded with manuscript text when they are mentioned.

6.       Lines 203-204: I'm not sure about this passage because it can cause confusion to the readership. Although the watermelon plants are seedlings, the symptoms of the disease caused by FON do not necessarily have to be non-visible or undetectable and even less so in a plant population with a suitable genetic pool after 30 days. I recommend rephrasing this sentence or supplementing it to provide the highest clarity for readers.

7.        Line 140: why were these treatments and experiments selected? Clear criteria are not mentioned.

8.       Figure 3: An better explanation of each subfigure is missing. For instance, what does each subfigure denoted by letters A to I mean? Each treatment? Specify it in the figure caption.

9.       Figures 2 and 3: The control is missing. In fact, each figure should be self-explainable without the need to check the main text for these details. The Y-axis is labeled as "Relative gene expression," so the control that was used to calculate the relative values should be provided in the figure caption. Finally, the full names of the tested transcripts must also be provided.

10.   Discussion should be improved since it is difficult to follow the main idea. I suggest subdividing the discussion into subheadings by the effect of each micronutrient and, in the end, providing a general discussion with the main or overall findings. This will be very helpful for readers.

11.   Revise in detail the M&M section. Some experimental details are missing to ensure outcome reproducibility. For instance, the brand, model, and grade of reagents, solvents, materials, and instruments must be provided. In addition,

12.   Line 447: Since the authors used 2^-ddCT for gene expressions, this method requires the authors to define a control tissue, where the expression will be relative to it, and also determine the reference genes (housekeepers) that were used to normalize the expression. The authors mention that they used ß-Actin, but nothing about standardization is mentioned. Even the sequence and the exact name of the control must be provided. In fact, any information on these terms (housekeeping gene and control tissue) is provided in the whole article. For instance, Table 2 and figures 2-3 do not have primers used for reference genes, and the subsection in line 447 does not describe the tissue selected to calculate the relative expression.

13.   Line 452: More details of collected leaves must be provided, such as status, plant characteristics, plant stratum where leaf samples were collected, etc.

14.   Conclusions section is missing.

Author Response

(The authors gave the same response as above.)

Reviewer 3 Report

  This study has little practical value and  low theoretical significance. 

No significant phenotypic differences of Watermelon Fusarium Wilt or RKN galling were observed after micronutrient treatmentsso it is meaningless to study the expression of resistance genes induced by micronutrient.

Author Response

We agree that significant phenotypic differences were not observed with respect to the severity of wilt symptoms or RKN galling. However, we humbly disagree with the reviewer’s comment on the practical value and significance of this piece of research.

As stated in the manuscript, previous works on soil-based mediums can identify the integrated and cumulative effect of micronutrients on induced resistance in response to plant disease, but those methods cannot identify the role of individual micronutrients. In this study, we used the hydroponic system to overcome the limitation of previous soil-based methods to study the role of individual micronutrients in induced resistance to plant disease. Although no significant phenotypic differences were observed with respect to the severity of wilt symptoms or RKN galling with any of the micro-nutrient treatments within the 30 day-experimental-period, the following differences were observed:

  1. the level of IR gene expression varied with sampling period, type and concentration of micro-nutrients applied, and pathogen-inoculation.
  2. In the absence of pathogens, no significant changes were observed in the expression level of IR genes on the 7th day of micronutrient treatment.
  3. Pathogen inoculation affected the expression levels of the IR genes at 3-day post-inoculation.
  4. In FON inoculated plants, PDF was upregulated in high Fe treatment, whereas in RKN inoculated plants, low Mn treatment resulted in upregulation of VSP.
  5. In the case of mixed inoculation with FON and RKN, the plants with low Zn treatment resulted in the upregulation of PR1.

These observations suggest that the type and concentration of micronutrients in watermelon may potentially induce systemic resistance against FON and RKN through SA and JA pathways. These findings are of significant impact and will serve as the foundation for future studies on the role of individual micronutrients on global gene expression profiles before and after the onset of disease in plants.

Round 2

Reviewer 2 Report

Authors addressed adequately my comments. Manuscript improved in quality and content. Therefore, I recommend its acceptance.